# The Curious Case of Nonverbal Abstract Reasoning with Multi-Modal Large Language Models

**Kian Ahrabian**,* **Zhivar Sourati**\*, **Kexuan Sun**\*, **Jiarui Zhang, Yifan Jiang,**
**Fred Morstatter & Jay Pujara**
Information Sciences Institute
University of Southern California
Marina Del Rey, CA 90292, USA
`{ahrabian,souratih,kexuansu,jzhang37,yjiang44}@usc.edu`
`{fredmors,jpujara}@isi.edu`

## Abstract

While large language models (LLMs) are still being adopted to new do-
mains and utilized in novel applications, we are experiencing an influx
of the new generation of foundation models, namely multi-modal large
language models (MLLMs). These models integrate verbal and visual infor-
mation, opening new possibilities to demonstrate more complex reasoning
abilities at the intersection of the two modalities. However, despite the
revolutionizing prospect of MLLMs, our understanding of their reasoning
abilities is limited. In this study, we assess the nonverbal abstract reason-
ing abilities of open-source and closed-source MLLMs using variations
of Raven's Progressive Matrices. Our experiments reveal the challenging
nature of such problems for MLLMs while showcasing the immense gap
between open-source and closed-source models. We also uncover critical
shortcomings of visual and textual perceptions, subjecting the models to
low-performance ceilings. Finally, to improve MLLMs' performance, we
experiment with different methods, such as Chain-of-Thought prompting,
leading to a significant (up to 100%) boost in performance. Our code and
datasets are available at https://github.com/usc-isi-i2/isi-mmlm-rpm.

## 1 Introduction

Foundation models — mostly large language models (LLMs) and large vision models (LVMs)
— have revolutionized the field of artificial intelligence, demonstrating zero-shot (Radford
et al., 2019; Wang et al., 2023a) and few-shot (*i.e.,* in-context) learning abilities (Brown et al.,
2020; Zhang et al., 2023b) that perform on-par or even surpass humans in some tasks (Webb
et al., 2023). These tasks cover both dimensions of general intelligence: crystallized in-
telligence focusing on retrieving knowledge from memory (Hartmann et al., 2023) and
fluid intelligence involving novel and abstract reasoning (Cattell, 1987). Following these
advancements, there has been a recent surge in the development of a new generation of
foundation models, namely, multi-modal large language models (MLLMs). These models
can process visual and textual cues (OpenAI, 2023; Zhao et al., 2023; Huang et al., 2023),
paving the way for solving far more complex tasks concerning both modalities.

Nonverbal abstract reasoning is a family of tasks that involve both modalities. It has
been studied extensively for measuring fluid intelligence (Shakeel et al., 2017), in which
reasoners need to demonstrate strong visual perception and high-level explicit reasoning
abilities to solve these tasks. Prior studies have explored the performance of LVMs (Zhuo &
Kankanhalli, 2020) and LLMs (Hu et al., 2023; Webb et al., 2023) on transformed versions
of these tasks in a uni-modal setting. However, theoretical and empirical evidence exists
for the benefits of the interplay between verbal and visual perceptions (Vyshedskiy, 2019;
Winawer et al., 2007; Vygotsky, 1962), suggesting that visual perception helps us better

---

\* Authors contributed equally.

understand our surroundings, while language helps us symbolize and facilitate reasoning through notions such as self-talk (Berk, 1994).

Inspired by prior works (Vygotsky, 1962; Lupyan, 2012; Nam et al., 2017; Colas et al., 2021) that have explored the fusion of visual and verbal cues and their effect on cognition and reasoning abilities, and considering the opportunity of experimenting with both modalities in MLLMs, in this study, we strive to answer the following research question: "**Do MLLMs demonstrate faithful nonverbal abstract reasoning abilities?**"

Our contributions are as follows:

1. We evaluate the nonverbal abstract reasoning abilities of 24 open-source and closed-source MLLMs under three Raven's Progressive Matrices (RPM) (Raven, 2003) benchmarks (See Figure 1 for an example);

2. We evaluate MLLMs' textual and visual abilities in semi-isolated settings that mitigate cross-modality contamination, providing insights into their performance ceiling.

3. We evaluate MLLMs' zero-shot and few-shot abilities, drawing a more accurate picture of the alignment between their verbal and visual perceptions;

All in all, we observe that while open-source MLLMs perform poorly on nonverbal abstract reasoning tasks, closed-source models such as GPT-4V (OpenAI, 2023) showcase non-trivial abilities (See Section 4). Moreover, we discover critical shortcomings in both visual and verbal capabilities across open-source and closed-source models, partially explaining the observed poor performances (See Section 5). Finally, we find closed-source models' textual and visual perceptions to be relatively aligned, allowing us to improve their performance significantly by providing guided prompts and in-context demonstrations (See Section 6).

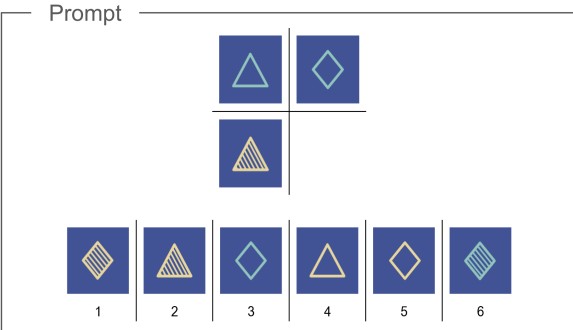

Figure 1: An example of model's prediction on a sample from the IQ50 dataset. Given a prompt with a visual puzzle (*top*), the model generates a response that includes its reasoning and the chosen option.

## 2 Related Work

**Foundation Models' Reasoning Abilities.** With the advancements of large pre-trained (*i.e.*, foundation) models (Vaswani et al., 2017; Dosovitskiy et al., 2020), researchers have extensively evaluated their reasoning abilities (Bubeck et al., 2023). These evaluations go beyond simple knowledge retrieval (Zhu et al., 2023b; Hartmann et al., 2023), focusing on tasks that require novelty and abstractions built on models' knowledge (Rytting & Wingate, 2021), covering visual (Zhuo & Kankanhalli, 2020; Jahrens & Martinetz, 2020; Barrett et al., 2018) and textual (Webb et al., 2023; Hu et al., 2023; Lu et al., 2022; Hill et al., 2019; Wei et al., 2022) dimensions.

**MLLMs.** Wide-spread utilization of foundation models and their role as general-purpose interfaces for different modalities (Hao et al., 2022) have led to the development of

MLLMs (Li et al., 2022; Chen et al., 2022; Alayrac et al., 2022; Li et al., 2023a; Wang et al., 2022; Zhu et al., 2023a) that can generate text conditioned on the combination of different modalities, demonstrating zero-shot (Li et al., 2022), few-shot (Alayrac et al., 2022; Zhao et al., 2023), and chain-of-thought abilities (Huang et al., 2023). To better understand their range of abilities, prior studies have evaluated MLLMs for geometric understanding and reasoning (Kazemi et al., 2023), text recognition (Liu et al., 2023b), mathematical reasoning (Lu et al., 2024), college-level deliberate reasoning (Yue et al., 2023), and open-ended reasoning (Han et al., 2023) abilities. Most similar to our study are the works of Qi et al. (2023) and Mitchell et al. (2023), evaluating the abstract reasoning abilities of MLLMs; however, their evaluation is either limited to a few examples or lack the rigor to provide an in-depth understanding. In this study, we bridge the gap in the literature by conducting extensive experiments that produce comprehensive insights into MLLMs' abstract nonverbal reasoning abilities.

# 3 Experimental Setup

## 3.1 Datasets

**IQ50.** Introduced by Huang et al. (2023), IQ50 is a nonverbal reasoning benchmark containing 50 visual puzzles crawled from the internet. Given a set of images arranged in a matrix or a sequence, the goal is to predict the missing piece of the puzzle from six given options. To conduct our extensive experiments, we augment each puzzle with textual description and hint annotations to explore the interplay between textual cues and visual perceptions.

**RAVEN.** Introduced by Zhang et al. (2019), RAVEN is a visual reasoning dataset containing 70,000 synthetic samples in seven categories (each 10,000). Each instance is created following a sampled rule and contains a 3x3 matrix of images (with the bottom right piece missing) and eight options, with a goal similar to IQ50. To reduce the computational costs, we randomly sample 500 examples from each category (3,500 in total) to create RAVEN-S.

**CCSE.** We collected 175 visual abstractions and reasoning problems from the China Civil Service Examination (CCSE), each containing a matrix or a sequence of images with four options. This newly curated dataset serves as a challenging benchmark across various reasoning patterns. See Appendix F for more details.

## 3.2 Models

**Pre-Trained.** The first set of models that we utilize in our experiments are state-of-the-art pre-trained MLLMs. Specifically, we chose the following models due to their popularity and accessibility: 1) BLIP-2 (Li et al., 2023a), 2) Fuyu (Bavishi et al., 2023), 3) IDEFICS (Laurençon et al., 2023), and 4) Qwen-VL (Bai et al., 2023). Note that all these models, except for the BLIP-2 family, have undergone a multi-task pre-training procedure, presumably allowing them to have zero-shot abilities on a wide range of tasks (See Appendix B).

**Instruction-Tuned.** The second set of models that we use in our experiments are state-of-the-art instruction-tuned MLLMs. Specifically, we chose the following models with similar criteria as pre-trained MLLMs: 1) InstructBLIP (Dai et al., 2023), 2) MMICL (Zhao et al., 2023), 3) LLaVA (Liu et al., 2023a) 4) IDEFICS (Laurençon et al., 2023), 5) Qwen-VL (Bai et al., 2023), 6) GPT-4V (OpenAI, 2023), and 7) Gemini (Google, 2023).

**Heuristics.** Since most of our samples follow similar spatial patterns and the task is to fill the missing image with the best candidate, we can write the expected representation of the target image as a function of the provided query images. To this end, we first compute the target image's expected representation as

$$r = \sum_{q \in Q} \alpha_q R(q), \tag{1}$$

where $R$ is a function mapping images to vector representations, $Q$ is the set of all query images, and $\alpha_q \in \mathbb{R}$ is the weight of image $q$. For simplicity, we only consider linear

combinations. Then, we select the candidate $p$ that has the highest similarity to $r$ as our prediction:

$$p = \underset{c \in C}{\text{argmax}} \, S(r, R(c)), \tag{2}$$

where $C$ is the set of all candidate images, and $S$ is the Euclidean distance. For example, in Figure 1, a heuristic could be formulated as:

$$q_{12} - q_{11} = q_{22} - q_{21} \implies q_{22} = q_{21} + q_{12} - q_{11}, \tag{3}$$

which leads to $\alpha_{q_{11}} = -1$, $\alpha_{q_{12}} = 1$ and $\alpha_{q_{21}} = 1$. See Appendix C for details on selecting $R$ and calculating $\alpha_q$.

**Control Baselines.** We also report the random and majority-based performance for all the datasets as control baselines, enabling us to put the posted performances in perspective.

### 3.3 Implementation Details

We use greedy decoding (*i.e.,* no sampling) with temperature $= 0.0$ and top_p $= 1.0$ for all the tested models. Moreover, max_generation_length is set to 512 across all experiments. Furthermore, for gpt-4v, we set the model's resolution to auto. All our experiments are carried out on a server with 4 × Quadro RTX 8000 GPUs with 48GB VRAM, 251GB RAM, and 32 CPU cores. Finally, we implemented our code using Hugging Face Transformers (Wolf et al., 2020) and PyTorch (Paszke et al., 2019) libraries.

## 4 How good are MLLMs at nonverbal abstract reasoning?

### 4.1 Automatic Scoring

Our early observations of the MLLMs' responses indicated the possibility of option markers (i.e., 1, 2, etc.) appearing at different token positions. For example, if the model generates *"Number 4."*, the marker will be at position 3; however, if the model generates *"The answer is 4."*, the marker will be at position 7. As such, we pivoted away from the common next-token scoring approach and used a pattern-matching scheme. Specifically, we find all the number tokens in the response and then take the one with the largest logit as the chosen option to be compared to the ground truth. We further discuss this approach in Appendix A, making a comparison to another common automatic scoring method.

| Model | IQ50 | RAVEN-S | CCSE |
|---|---|---|---|
| **Pre-Trained** | | | |
| blip2-opt-2.7b | 0.160 | 0.122 | 0.194 |
| blip2-opt-6.7b | 0.140 | 0.117 | 0.229 |
| blip2-flan-t5-xl | 0.160 | 0.117 | 0.229 |
| blip2-flan-t5-xxl | 0.180‡ | **0.131**‡ | 0.211 |
| idefics-9b | 0.120 | 0.120 | 0.194 |
| idefics-80b* | **0.240**‡ | T | 0.240 |
| fuyu-8b | 0.160 | 0.127‡ | 0.297‡ |
| Qwen-VL | 0.180‡ | 0.117 | 0.206 |
| **Instruction-Tuned** | | | |
| MMICL-vicuna-7b | 0.200‡ | 0.115 | 0.257‡ |
| MMICL-vicuna-13b* | 0.180‡ | 0.126‡ | 0.223 |
| MMICL-Instructblip-T5-xl | 0.160 | 0.126‡ | 0.229 |
| MMICL-Instructblip-T5-xxl | 0.200‡ | 0.126‡ | 0.229 |
| instructblip-vicuna-7b | 0.140 | 0.126‡ | 0.240 |
| instructblip-vicuna-13b* | 0.160 | 0.117 | 0.217 |
| instructblip-flan-t5-xl | 0.120 | 0.121 | 0.240 |
| instructblip-flan-t5-xxl | **0.240**‡ | 0.126‡ | 0.211 |
| idefics-9b-instruct | 0.120 | 0.121 | 0.217 |
| idefics-80b-instruct* | 0.140 | T | 0.251‡ |
| llava-1.5-7b-hf | 0.160 | 0.123 | 0.269‡ |
| llava-1.5-13b-hf* | **0.240**‡ | 0.121 | 0.229 |
| bakLlava-v1-hf | 0.080 | 0.122 | **0.314**‡ |
| Qwen-VL-Chat | 0.220‡ | 0.117 | 0.286‡ |
| **Heuristics** | | | |
| Pixel | 0.200 | 0.051 | 0.257 |
| CLIP-ViT | 0.480 | 0.099 | 0.234 |
| **Control Baselines** | | | |
| Random | 0.167 | 0.125 | 0.250 |
| Majority | 0.220 | 0.130 | 0.314 |

Table 1: Zero-shot accuracy on IQ50, RAVEN-S, and CCSE datasets using the generalized next-token scoring method. For each dataset, the best performance by MLLMs is **boldfaced** while the second best performance is underlined. **Legends:** T → Timeout after one week of running, * → Ran with half-precision (e.g., bfloat16) to fit in GPU memory, ‡ → Performance better than the random baseline.

Table 1 presents our experimental results on IQ50, RAVEN-S, and CCSE datasets using our automatic scoring method. Our main observation from this table is that **no model**

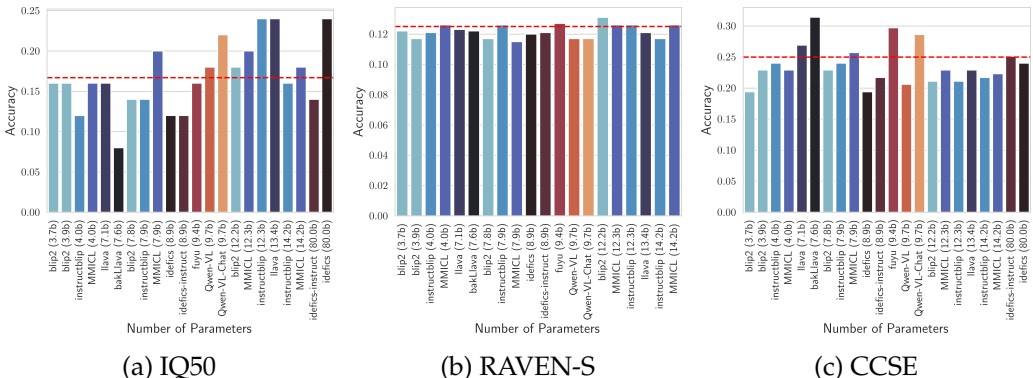

Figure 2: Zero-shot accuracy concerning the number of parameters using the automatic scoring method. Models are sorted from smallest (left) to largest (right), and those within the same family are colored the same. The red dashed lines indicate the random baselines.

**consistently beats the random baselines over the three datasets**. Moreover, compared to the random baselines, the models perform within the $[-8.7\%, +7.3\%]$, $[-1.0\%, +0.6\%]$, and $[-5.6\%, +6.4\%]$ ranges for IQ50, RAVEN-S, and CCSE, respectively. Although some models achieve non-trivial gains over the random baselines and even outperform some of the heuristic baselines, these results expose the difficulty of solving variations of the RPM-style questions across a wide range of MLLMs.

Comparing the best-achieved performances with the majority baselines, we observe a stark similarity ($\pm 2\%$). This begs the question of whether the posted numbers are true representatives of the reasoning powers of MLLMs or simply a side-effect of their generation biases (*e.g.,* a model that always generates 1s will do very well on a dataset with many 1s as gold labels). To this end, in Section 4.2, we manually examined the generated answers by the instruction-tuned models. Moreover, looking at the results posted by the pre-trained models, apart from `fuyu-8b`, we cannot observe a significant upside to the multi-task pre-trained models.

It is also worth noting that, by examining the results posted within and beyond the same family of models, we observe that the scaling law in terms of model size (Kaplan et al., 2020) (*i.e.,* the larger the model, the higher the performance) does not hold here. Figure 2 illustrates the models' zero-shot accuracy concerning the number of parameters.

## 4.2 Manual Scoring

One critical aspect of assessing such models' abilities is ensuring the correctness and faithfulness of their reasoning. As such, we manually inspect the generated responses to provide insights into the results posted by the models in the automatic scoring schemes. All the inspections were done by a group of three graduate students (See Appendix D for the rubric). Moreover, to elicit reasoning, we appended the phrase "*Let's think step by step.*" to the prompt (Kojima et al., 2022). Out of the 14 open-source instruction-tuned mod-

| Model | ✓A ✓R | ✗A ✓R | ✓A ✗R |
|---|---|---|---|
| gpt-4v | **0.26** | **0.16** | 0.10 |
| gemini-pro-vision | 0.10 | 0.14 | **0.16** |
| llava-1.5-7b-hf | 0.00 | 0.00 | 0.02 |
| llava-1.5-13b-hf | 0.04 | 0.02 | 0.10 |
| idefics-9b-instruct | 0.00 | 0.00 | 0.08 |
| Qwen-VL-Chat | 0.00 | 0.00 | 0.04 |

Table 2: Performances of instruction-tuned models on IQ50, assessed by manual inspection, in terms of **A**nswer and **R**easoning correctness, indicated by **A** and **R**, respectively. The best performance is **boldfaced** while the second best performance is underlined.

els (See Table 3), we were only able to get meaningful and coherent responses from the following models: `llava-1.5-7b-hf`, `llava-1.5-13b-hf`, `Qwen-VL-Chat`, and `idefics-9b-instruct`. Besides these models, we also manually measured the abstract reasoning abilities of closed-source MLLMs (*i.e.,* `gpt-4v` and `gemini-pro-vision`), as we don't need access to the gen-

erated logits anymore. For the remainder of our experiments, we relied on IQ50, a small, challenging test set that is easy for humans to solve (see Appendix E), based on our preliminary tests, but difficult for MLLMs (See Table 1). This choice also allows us to expand our study more efficiently with various experiments.

Table 2 presents the performance of the aforementioned models on IQ50, demonstrating their poor explicit reasoning capabilities. More specifically, only one of the open-source instruction-tuned models achieves a non-zero performance on the joint answer and reasoning correctness, with an abysmal performance of 4%, (*i.e.,* an average of 1% across open-source models). Meanwhile, the trend in the closed-source MLLMs is more promising, with `gpt-4v` outperforming random and majority baseline, providing correct reasoning and answers in 26% of the samples. Nevertheless, their performance still lags behind the simple heuristics by a large margin (See Table 1). Regarding the faithfulness of answers to reasonings, we only observe a meaningful level of faithfulness (alignment between reasoning and answer when either is correct) in closed-source models, 50% for `gpt-4v` and 25% for `gemini-pro-vision`. Observing such a high percentage of unfaithfulness to reasoning further asserts the importance of conducting manual evaluations rather than merely reporting performances using noisy automatic measurements.

During our inspections, we observed that although models generally understand the shapes in the query images, they tend to hallucinate about specific features such as rotation, shadows, and orientations. On the reasoning side, the most prominent issue was the models being overly descriptive rather than being focused on providing grounded logical reasoning. In other words, most of the time, the generated responses were the descriptions of the puzzle and the candidate options. We believe this to be an artifact of their training datasets; as such, we expected a much better showing from the models that receive multi-task training (See Table 3).

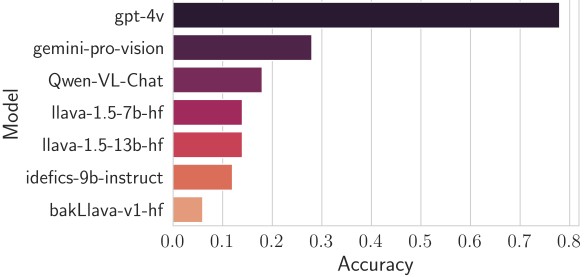

Figure 3: Zero-shot CoT accuracy on IQ50 using text-only prompts.

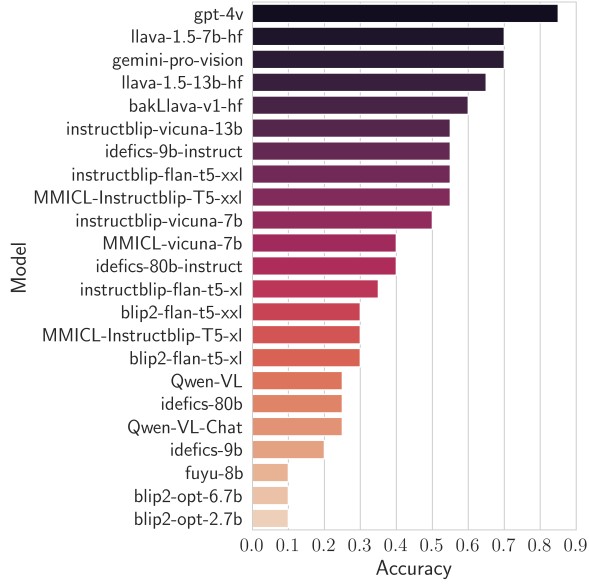

Figure 4: Visual awareness questions performances on a subset of IQ50.

## 5   What are the root causes of the poor performance?

**Textual Reasoning.** Since most MLLMs fuse the information from visual and textual modalities, they are prone to error propagation from each. Hence, if we bypass one module, we can effectively evaluate the remaining one. For the textual module, we can do this by providing the text-only version of each sample (*i.e.*, the textual description of the puzzles) written by a human expert (See Appendix G). Through these experiments, we can gain insights into the ceiling reasoning capabilities of each model. In our experiments, due to their incompatibility with text-only prompts, we had to drop `MMICL*` and `InstructBLIP*` models. Moreover, we dropped `idefics-80b-instruct` as we could not elicit reasonings from it.

Figure 3 presents a comparison between open-source and closed-source MLLMs on IQ50 using text-only zero-shot CoT prompts. As evident, `gpt-4v` is the only model that achieves a high level of performance, with `gemini-pro-vision` coming in as a distant second. These results are consistent with our previous observations, where open-source models struggled to achieve good results, showcasing a lag in textual reasoning abilities.

**Visual Awareness.** Correctly perceiving visual details is critical to nonverbal abstract reasoning. However, Zhang et al. (2023a) have shown that MLLMs face difficulties when isolating granular details in large images. As such, we ran a set of experiments to determine the extent to which our tested MLLMs understand the presented puzzles. More concretely, we developed 20 questions designed to test the models on understanding shape, relative position, orientation, color, filling pattern, and fine-grained details. In our experiments, we dropped `MMICL-vicuna-13b` as we could not elicit proper responses.

Figure 4 presents the performances posted on the visual awareness questions across open-source and closed-source models. As evident, `gpt-4v` dominates the benchmark with a comfortable lead; however, we find it very promising that some open-source models such as `llava*` can keep up with the other closed-source model (*i.e.*, `gemini-pro-vision`). Overall, observing the low performances of open-source MLLMs in visual awareness and textual reasoning paints a clearer picture of models' shortcomings, explaining their results in Section 4.

# 6 Can MLLMs' performance be improved?

## 6.1 Guided Prompting

Prompt engineering has been one of the prominent methods to guide LLMs towards specific desired outputs through better conditioning (Wei et al., 2022; Kojima et al., 2022; Wang et al., 2023b; Li et al., 2023b). We experiment with three guided prompting setups, each providing the models with different cues to understand how supplementary textual information is utilized while generating responses. The three setups are as follows (See Appendix G for examples):

- **General.** In this setup, we provide the models with broad cues on approaching such visual puzzles, hinting at the common strategies without being sample-specific.

- **Sample-specific.** In this setup, we provide the models with one sample-specific hint about the desired reasoning for solving each puzzle.

- **Corrective.** In this setup, in an interactive process, looking at the model's reasoning when prompted in a zero-shot setting, we add one hint to correct the most prominent error.

Figure 5 illustrates the performance of closed-source models on IQ50 given different hints. Looking at these results, we find sample-specific and general hints detrimental to reasoning and accuracy while observing a significant boost when interactively utilizing corrective hints, especially for `gpt-4v`. Considering the distinct embedded cues of sample-specific and corrective hints, we believe that the models' inherent chain of reasoning on the provided puzzles is misaligned with humans, which makes sample-specific hints confusing rather than helpful. However, we can correct models' already laid-out solutions with corrective hints, improving their performance by as much as 100%.

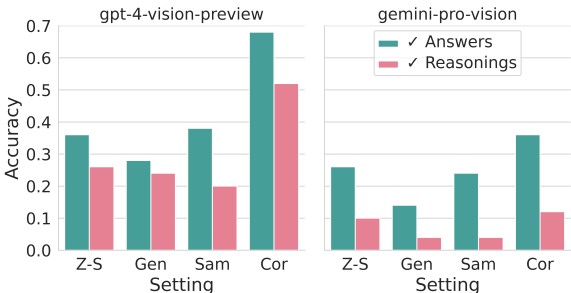

Figure 5: Guided prompting performance of `gpt-4v` and `gemini-pro-vision` on IQ50 using different types of hints. **Legend:** Z-S → Zero-shot, Gen → General, Sam → Sample-specific, and Cor → Corrective.

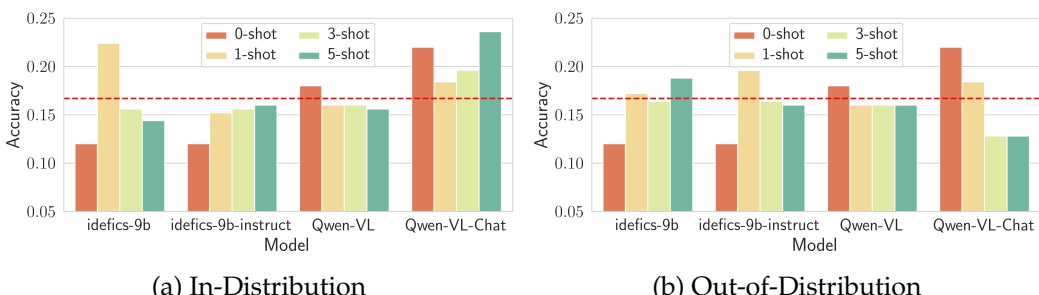

(a) In-Distribution        (b) Out-of-Distribution

Figure 6: Zero-shot and symmetrical few-shot accuracy on IQ50. In *(a) In-Distribution*, the demonstrations are taken from IQ50, while in *(b) Out-of-Distribution*, the demonstrations are taken from RAVEN-S. Each variation was executed five times with different seeds to mitigate the effect of random sampling. The red dashed lines indicate the random baselines.

## 6.2 In-Context Learning

In-context learning (ICL) refers to emergent behavior in LLMs where they perform a task conditioned on the provided demonstrations without further parameter optimization. Min et al. (2022) have suggested that ICL is a mere mechanism for locating the already learned ability of the model to respond to the query prompt. However, more recent studies have shown that LLMs can learn various function classes through ICL (Garg et al., 2022; Mirchandani et al., 2023; Lee et al., 2023).

### 6.2.1 Symmetrical Few-Shot

The most common form of ICL is few-shot, in which the model is provided with demonstrations similar to the test sample. We call this variation "Symmetrical" as there is no imbalance in the demonstrations' textual and visual information pieces. Since symmetrical ICL requires processing multiple image and text pairs, we only utilize models capable of processing such inputs (See Table 3 for a list). Moreover, due to burdensome computation costs, we exclude `idefics-80b*` models. In our experiments, we explored the effect of 1) changing the sampling distribution of demonstrations and 2) including step-by-step reasoning with demonstrations (See Appendix G for examples).

**Effect of sampling distribution.** Our experiments cover open-source models and utilize our automatic scoring scheme over two variations: 1) *In-Distribution (ID):* demonstrations are uniformly taken from the same dataset, and 2) *Out-of-Distribution (OOD):* demonstrations are uniformly taken from another dataset. We used IQ50 as the evaluation source with RAVEN-S as the OOD source. Moreover, we ran each variation five times to reduce the impact of random sampling and up to 5-shot (*i.e.,* 10% of the dataset) due to GPU limits.

Figure 6 presents the results of our experiments. As evident, there are no consistent patterns across models and variations. For example, while `idefics-9b*` models generally benefit from the few-shot scheme, we don't see monotonically increasing performances with more demonstrations, contrary to expected patterns in LLMs. Simultaneously, we see a consistent decline in performance in some variations of `Qwen-VL*` models. We believe these irregularities are caused by an inability to understand the utility of the demonstrations, suggesting that the tested models do not exhibit strong and consistent symmetrical ICL capabilities. As a result, the models cannot

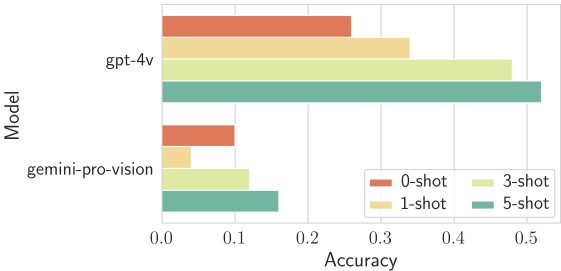

Figure 7: Symmetrical few-shot CoT accuracy on IQ50.

take advantage of the provided demonstrations properly, leading to poor responses and subpar performances.

**Effect of step-by-step reasoning.** Inspired by its success, we experiment with the few-shot Chain-of-Thought (CoT) prompting (Wei et al., 2022) to improve the performance of our models. Our early experiments found open-source models, such as `Qwen-VL-Chat` and `idefics-9b-instruct`, unable to comprehend CoT prompting. As such, we focused on examining the closed-source models: `gpt-4v` and `gemini-pro-vision`.

Figure 7 presents the performance of closed-source models with CoT prompting. We can observe that both models benefit significantly from CoT demonstrations, with `gpt-4v`'s performance being boosted as much as 100%. These results emphasize the immense gap between the open-source and closed-source models while showcasing meaningful symmetrical ICL abilities in these models. During our evaluation, we noticed an unusual phenomenon with `gemini-pro-vision` in which the one-shot variation prevented the model from generating reasonings in almost all the examples, leading to an initial performance drop. Moreover, in a similar setting for `gpt-4v`, we observed 1) a massive jump in the number of safeguard triggers (4% → 18%), which precluded the model from generating a response, and 2) an increased confusion regarding the boundaries of the provided demonstration (0% → 10%), diminishing the model's performance.

### 6.2.2 Asymmetrical Few-Shot

Adding new modalities has brought forth the possibility of providing lopsided (*i.e.,* Asymmetrical) information in the input prompt. As such, we conduct a set of few-shot experiments that provide the models with text-only CoT demonstrations while keeping the query unchanged (*i.e.,* image + text). To encourage the models to use the demonstrations, we append "Let's solve the puzzle in the image, step by step, similar to the demonstrations." to our prompt. We hypothesize that similar to the findings of Min et al. (2022), the models will better understand the input and output spaces and achieve improved performance.

Since we could not make open-source models properly utilize the textual demonstrations in our preliminary experiments, we continued our experiments only on closed-source models. Figure 8 illustrates the results of our experiments with asymmetrical few-shot CoT prompting. As evident, this approach does not yield meaningful improvements and even causes degradation in some cases, contrasting our hypothesis. Moreover, we noticed increased hallucinations, mostly confusing the demonstrations with queries and detecting non-existent details in the shapes, leading to unfaithfulness and instability in the responses. We leave further investigations and experiments on this ICL scheme to future works.

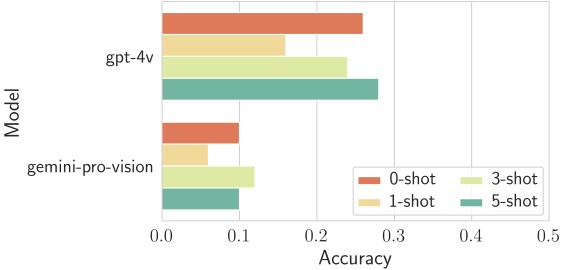

Figure 8: Asymmetrical few-shot CoT accuracy on IQ50.

## 7 Conclusion

In this study, we utilized different RPM-style tasks as a proxy for measuring the nonverbal abstract reasoning abilities of MLLMs, covering 24 different open-source and closed-source models. Although closed-source MLLMs showcased promising capabilities in our experiments, we found the abilities of open-source models to be insufficient for solving these tasks. Moreover, using pseudo-isolated experimental environments, we found that MLLMs often fail at 1) gathering precise visual details from puzzles and 2) reasoning correctly and faithfully, even when provided with expert-written and complete descriptions of the puzzles. Furthermore, our experiments highlighted 1) the inability of open-source models to consistently utilize demonstrations and 2) the ICL prowess of the closed-source models,

which helped them benefit from interactive guidance or provided demonstrations with step-by-step (*i.e.*, CoT) reasoning. Although MLLMs have previously demonstrated proficiency at various tasks, our study using a relatively simple reasoning task for humans has exposed some critical shortcomings in MLLMs while highlighting the importance of more grounded evaluations, even at small scales.

## Acknowledgements

This work has been sponsored and funded by the Defense Advanced Research Projects Agency via awards HR00112220046 and N660011924033, and contract HR00112390061. We thank Dong-Ho Lee, Pei Zhou, and Pegah Jandaghi for their invaluable feedback.

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

# A  One-by-One Scoring

Introduced by Huang et al. (2023), in this scoring method, we first flatten the query image matrix and feed it into the model along with exactly one option. We also surround these images with textual instructions to help the model better understand the desired task (See Appendix G for examples). Then, we calculate the probability of the model generating "Yes", representing the probability of that option being the true missing piece. Finally, we determine the model's choice by taking the option with the highest probability. To improve the original method, we introduce an equivalency condition in which the max probability of different variations of the target token (*e.g.,* "YES" and "yes") is taken as the probability. The main shortcomings of this method include 1) being more computation-heavy as it needs to pro-

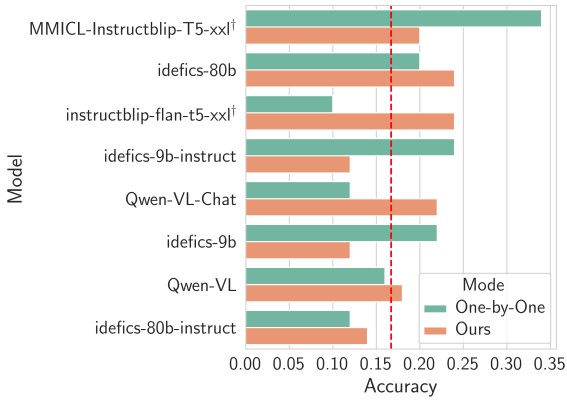

Figure 9: Zero-shot accuracy comparison on the IQ50 dataset using the one-by-one and our automatic scoring methods. Results with a † marker are taken from Zhao et al. (2023). Due to runtime errors, we could not replicate them (neither Huggingface nor GitHub versions). The red dashed line indicates the random baseline.

cess each option separately and 2) being compatible with only the models that accept multiple images as input. Notably, among the open-source models utilized in this study, only Qwen-VL* and idefics* models support multi-image inputs (See Table 3).

Figure 9 compares the experimental results with the one-by-one and our automatic scoring methods. As evident, some models such as MMICL-Instructblip-T5-xxl, idefics-9b, and idefics-9b-instruct benefit significantly from this scoring mode, while others such as instructblip-flan-t5-xxl and Qwen-VL-Chat suffer extensively. This observation demonstrates the validity of both methods, making the choice subject to the task/dataset being evaluated. However, given these mixed results and the downsides of using this scoring method, we conclude that our automatic scoring method is more practical for future studies.

# B  Models

Table 3 presents an attribute comparison over open-source and closed-source models.

**BLIP-2 (Li et al., 2023a).**  BLIP-2 is a generic and efficient pre-training strategy that bootstraps vision-language pre-training from off-the-shelf frozen pre-trained image encoders and frozen large language models. BLIP-2 bridges the modality gap with a lightweight Querying Transformer (Q-former), which is pre-trained in two stages. Despite having significantly fewer trainable parameters than existing methods, BLIP-2 achieves state-of-the-art performance on various vision-language tasks.

**Fuyu (Bavishi et al., 2023).**  Fuyu is a multi-modal text and image transformer trained by Adept AI. Architecturally, Fuyu is a vanilla decoder-only transformer with no image encoder. Image patches are instead linearly projected into the first layer of the transformer, bypassing the embedding lookup. This simplification allows the model to support arbitrary image resolutions. Fuyu-8B improves over Qwen-VL on 2 out of the 3 most commonly used image-understanding datasets despite having 2B fewer parameters.

**IDEFICS (Laurençon et al., 2023).**  IDEFICS (Image-aware Decoder Enhanced à la Flamingo with Interleaved Cross-attentionS) is an open-access reproduction of Flamingo, a closed-source visual language model developed by Deepmind. It is built on top of two unimodal open-access pre-trained models with newly initialized parameters in the form of Transformer

| Model | Size | Open Source | Multi-Task Pre-Training | Multi-Image Input |
|---|---|---|---|---|
| **Pre-Trained** | | | | |
| `blip2-opt-2.7b` | 3.7b | ✓ | ✗ | ✗ |
| `blip2-opt-6.7b` | 7.8b | ✓ | ✗ | ✗ |
| `blip2-flan-t5-xl` | 3.9b | ✓ | ✗ | ✗ |
| `blip2-flan-t5-xxl` | 12.2b | ✓ | ✗ | ✗ |
| `idefics-9b` | 8.9b | ✓ | ✗ | ✓ |
| `idefics-80b` | 80.0b | ✓ | ✗ | ✓ |
| `fuyu-8b` | 9.4b | ✓ | ✓ | ✗ |
| `Qwen-VL` | 9.7b | ✓ | ✓ | ✓ |
| **Instruction-Tuned** | | | | |
| `gpt-4-vision-preview` | U | ✗ | U | ✓ |
| `Bard (Gemini Update)` | U | ✗ | U | ✓ |
| `MMICL-vicuna-7b` | 7.9b | ✓ | ✓ | ✓* |
| `MMICL-vicuna-13b` | 14.2b | ✓ | ✓ | ✓* |
| `MMICL-Instructblip-T5-xl` | 4.0b | ✓ | ✓ | ✓* |
| `MMICL-Instructblip-T5-xxl` | 12.3b | ✓ | ✓ | ✓* |
| `instructblip-vicuna-7b` | 7.9b | ✓ | ✓ | ✗ |
| `instructblip-vicuna-13b` | 14.2b | ✓ | ✓ | ✗ |
| `instructblip-flan-t5-xl` | 4.0b | ✓ | ✓ | ✗ |
| `instructblip-flan-t5-xxl` | 12.3b | ✓ | ✓ | ✗ |
| `idefics-9b-instruct` | 8.9b | ✓ | ✗ | ✓ |
| `idefics-80b-instruct` | 80.0b | ✓ | ✗ | ✓ |
| `llava-1.5-7b-hf` | 7.1b | ✓ | ✓ | ✗ |
| `llava-1.5-13b-hf` | 13.4b | ✓ | ✓ | ✗ |
| `bakLlava-v1-hf` | 7.6b | ✓ | U | ✗ |
| `Qwen-VL-Chat` | 9.7b | ✓ | ✓ | ✓ |

Table 3: Comparison of open-source and closed-source models' attributes. **Legend:** U → Undisclosed, ∗ → We could not utilize this feature using the official code.

blocks to bridge the gap between the vision encoder and the language model. The model is trained on image-text pairs and unstructured multi-modal web documents. IDEFICS-instruct is the model obtained by further training IDEFICS on Supervised Fine-Tuning and Instruction Fine-Tuning datasets. IDEFICS is on par with the original closed-source model on various image-text benchmarks, including visual question answering (open-ended and multiple choice), image captioning, and image classification when evaluated with in-context few-shot learning.

**Qwen-VL (Bai et al., 2023).** Qwen-VL models are large-scale vision-language models (LVLMs) designed to perceive and understand texts and images. Starting from the Qwen-LM as a foundation, the model is endowed with visual capacity by the meticulously designed (i) visual receptor, (ii) input-output interface, (iii) 3-stage training pipeline, and (iv) multilingual multi-modal cleaned corpus. The resulting models, including Qwen-VL and Qwen-VL-Chat, set new records for generalist models under similar model scales.

**InstructBLIP (Dai et al., 2023).** InstructBLIP models are instruction-tuned MLLMs based on the pre-trained BLIP-2 models. They have been trained using 13 publicly available datasets transformed into instruction tuning format. Additionally, the authors introduce an instruction-aware Query Transformer, which extracts informative features tailored to the given instruction. Trained on 13 held-in datasets, InstructBLIP attains state-of-the-art zero-shot performance across all 13 held-out datasets, substantially outperforming BLIP-2 and larger Flamingo models.

**LLaVA-1.5 (Liu et al., 2024).** LLaVA-1.5 achieves state-of-the-art performance on various multimodal benchmarks through increased input resolution, additional layers of multimodal projection, and a comprehensively curated visual instruction tuning dataset. The model's efficient and lightweight architecture facilitates high reproducibility, positioning it as a versatile foundation for further research and development in the multimodal community.

**MMICL (Zhao et al., 2023).** Unlike previous work, MMICL utilizes a novel context scheme, treating image and text representations equally and establishing the reference between image and text via image declaration. It enables users to have the flexibility to input multiple images and text in any desired order, with no restrictions on the quantity or placement of images in contexts. MMICL achieves new state-of-the-art zero-shot performance on a wide range of general vision-language tasks, especially for complex benchmarks, including MME and MMBench. Moreover, MMICL effectively tackles the challenge of complex multi-modal prompt understanding and emerges with impressive ICL ability.

**GPT-4V (OpenAI, 2023).** GPT-4 with vision (GPT-4V) enables users to instruct GPT-4 to analyze image inputs provided by the user and is the latest capability OpenAI is making broadly available.

**Gemini (Google, 2023).** The Gemini family consists of Ultra, Pro, and Nano sizes, suitable for applications ranging from complex reasoning tasks to on-device memory-constrained use cases. Evaluation on a broad range of benchmarks shows that the Gemini Ultra model advances the state of the art in 30 of 32 of these benchmarks - notably being the first model to achieve human-expert performance on the well-studied exam benchmark MMLU and improving the state-of-the-art in every one of the 20 multi-modal benchmarks the authors examined. At the time of this publication, only the Pro version was publicly available.

## C   Heuristics Details.

### C.1   Selecting $R$

**Pixel.** Since raw pixel values are inputs to all MLLMs in this study, we utilize the flattened pixel values as the first variation of $R$ in our heuristics.

**CLIP-ViT (Dosovitskiy et al., 2021; Radford et al., 2021).** Contrasting Language-Image Pre-Training (CLIP) is adopted by most of the open-sourced MLLMs as their visual encoder. As such, we select CLIP encoding as the second variation of $R$ in our heuristics.

### C.2   Calculating $\alpha_q$

We first identify all possible $2 \times 2$ submatrices in the $m \times n$ query matrix. For the horizontal axis, considering that each $2 \times 2$ pattern spans across two columns, the number of unique horizontal positions is $n - 1$. Similarly, $m - 1$ unique vertical positions are on the vertical axis. Therefore, the total number of distinct $2 \times 2$ patterns that can be extracted from an $m \times n$ matrix is $(m - 1) \times (n - 1)$. To determine the expected target representation, we first apply the formula $4 = 3 + 2 - 1$ to each submatrix; then, we average over all the calculated values. Finally, we choose the option with the smallest Euclidean distance (*i.e.,* $S$) to the expected target representation.

## D   Manual Inspection Rubric

Our main challenge was to overcome the nuances that appear in reasonings generated by LLMs. As such, the evaluators first met to 1) determine the correct reasoning paths for the samples in the dataset and 2) review a series of sample responses generated by the models (e.g., GPT-4v, Gemini, etc.) to determine the evaluation strategy. Based on the observations in this initial meeting, the evaluators decided to allow for extra/wrong details

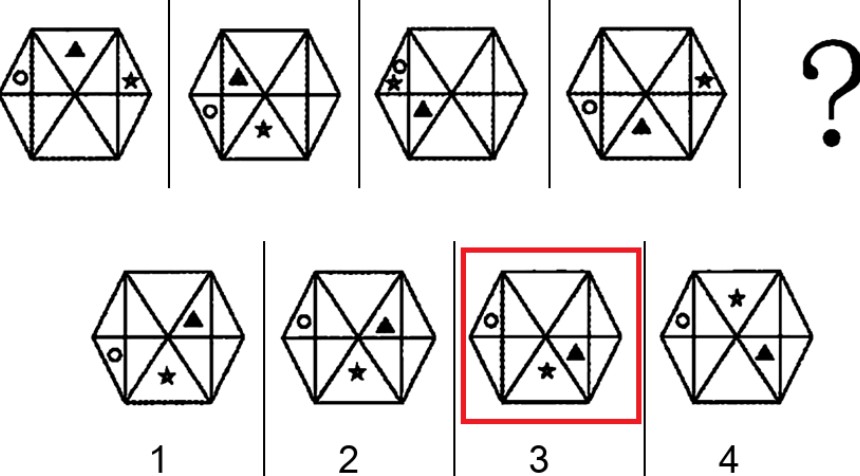

Figure 10: Example of the location pattern. Across the pieces, the triangle moves clockwise, one block at a time in the inner circle, while the five-pointed star moves clockwise, two blocks at a time in the outer circle.

in the generated responses as long as they did not affect or interfere with the alignment of the generated response to correct reasoning paths. For example, in some instances, the models perceived shadows (or incorrect colors) in the shapes that were not present in the puzzle; however, as long as they correctly detected the row-wise and column-wise change of patterns (e.g., square turning to circle) and grounded their reasoning on them, the evaluators marked the response as correct. Moreover, each sample was annotated and assessed by one person, and any uncertain case was flagged and shared among all three evaluators for discussion. After discussions, the final label was determined by a majority vote (i.e., at least 2 out of 3). All evaluators were mid-level to senior Computer Science PhD students specializing in NLP with extensive experience working with and evaluating LLMs.

While we acknowledge the difficulty of scaling such manual experiments, one of the main challenges of correctly assessing generated responses is that automatic metrics like ROUGE, BERTScore, etc., fall short of adequately evaluating the semantic nuances. Hence, we decided to go down the path of highly labor-intensive human expert evaluation to provide precise, concrete, and grounded insights.

## E  Human Performance on IQ50

To establish a baseline, we ran a study with 25 college-level participants from diverse educational and demographic backgrounds. We provided each participant with ten randomly selected samples from the IQ50 dataset (20% of the dataset) and instructed them to solve the puzzle and give a short reason for their answer. The average performance of this group was 95.9%, with a standard deviation of 6.92%.

## F  CCSE Dataset Examples

CCSE tests models' reasoning abilities over five general patterns (*i.e.,* location, logic, progression, self-geometry, and relative-geometry) in three types of figure configurations (*i.e.,* one-row, two-rows, matrix). See Figure 10 and Figure 11 for examples.

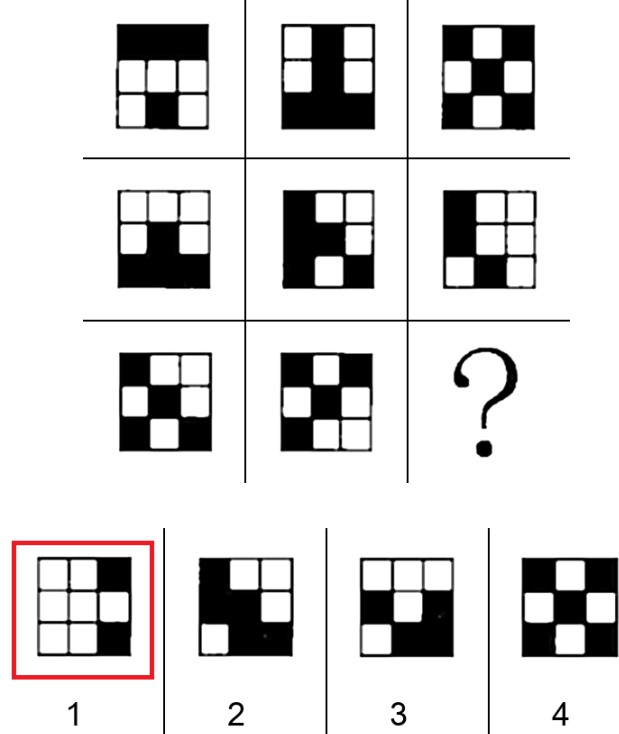

Figure 11: Example of the logic pattern. Each row follows a cell color XOR operation (0 for white and 1 for black) between the first and the second columns to make the piece in the third column.

## G   Prompt Examples

Table 4 and Table 5 present examples of each prompt used in our experiments.

## H   Limitations

In this work, we focused on only a specific type of reasoning task (*i.e.,* nonverbal abstract reasoning) while using only IQ50, a small but challenging dataset, in most of our experiments. However, despite these limitations, we hypothesize that similar shortcomings could be replicated in other reasoning tasks due to the fundamental and not task-specific nature of the observed problems. Moreover, although our experiments yielded insightful results, further analysis of the observations is still possible. For example, given the remarkable effectiveness of corrective hints, we can utilize methods such as self-talk Press et al. (2023), automating the whole process. Another example is the visual awareness tests, where it is possible to examine the internal values of the models (*e.g.,* attention weights, token probabilities, etc.), as opposed to evaluating the generated responses. Finally, since the closed-source models' training datasets are unknown, test set contamination is possible, leading to their superior performance. However, based on the relatively low performance of these models and considering the low-level difficulty of the tests for humans, we believe contamination to be unlikely.

| Experiment | Prompt Example |
|---|---|
| Automatic scoring | *[IMG]* You are given a puzzle. The puzzle features a set of visual patterns arranged in a matrix on the top, with the bottom right piece missing and six options at the bottom (marked by 1, 2, 3, 4, 5, or 6). Which option (either 1, 2, 3, 4, 5, or 6) fills the missing piece best? |
| One-by-one scoring | Here are three images: *[IMG1] [IMG2] [IMG3]* The following image is: *[IMG4]* Is it correct? |
| Zero-shot CoT | *[IMG]* You are given a puzzle. The puzzle features a set of visual patterns arranged in a matrix on the top, with the bottom right piece missing and six options at the bottom (marked by 1, 2, 3, 4, 5, or 6). Which option (either 1, 2, 3, 4, 5, or 6) fills the missing piece best? Let's think step by step. |
| Textual reasoning | Puzzle:
[[yellow percentage sign, yellow percentage sign],
[yellow percentage sign, ?]]
Options:
1: yellow percentage sign
2: yellow plus sign
3: two yellow circles
4: one yellow circle
5: yellow division sign
6: yellow cross
You are given a puzzle. The puzzle features a set of patterns arranged in a matrix on the top, with the bottom right piece missing and six options at the bottom (marked by 1, 2, 3, 4, 5, or 6). Which option (either 1, 2, 3, 4, 5, or 6) fills the missing piece best? Let's think step by step. |
| Visual awareness | In candidate 4 at the bottom, are the arrows arranged clockwise or counterclockwise? |
| Guided prompting (General) | *[IMG]* You are given a puzzle. The puzzle features a set of visual patterns arranged in a matrix on the top, with the bottom right piece missing and six options at the bottom (marked by 1, 2, 3, 4, 5, or 6). Which option (either 1, 2, 3, 4, 5, or 6) fills the missing piece best? Hint: Focus on the row-wise and column-wise changes regarding color, orientation, and shape of the puzzle pieces. Let's think step by step. |
| Guided prompting (Sample-specific) | *[IMG]* You are given a puzzle. The puzzle features a set of visual patterns arranged in a matrix on the top, with the bottom right piece missing and six options at the bottom (marked by 1, 2, 3, 4, 5, or 6). Which option (either 1, 2, 3, 4, 5, or 6) fills the missing piece best? Hint: the focus should be on the column-wise changes. Let's think step by step. |
| Guided prompting (Corrective) | **Turn 1.** *[IMG]* You are given a puzzle. The puzzle features a set of visual patterns arranged in a matrix on the top, with the bottom right piece missing and six options at the bottom (marked by 1, 2, 3, 4, 5, or 6). Which option (either 1, 2, 3, 4, 5, or 6) fills the missing piece best? Let's think step by step.
*[Model's Response]*
**Turn 2.** Hint: Option 1 does not have a small circle inside it, and option 5 is a very small circle itself. |
| Symmetrical few-shot | You are given a puzzle. The puzzle features a set of visual patterns arranged in a matrix on the top, with the bottom right piece missing and six options at the bottom (marked by 1, 2, 3, 4, 5, or 6). Which option (either 1, 2, 3, 4, 5, or 6) fills the missing piece best? Let's think step by step.
*[IMG1]*
The answer is 4.
*[IMG2]*
The answer is 1.
*[IMG3]* |

Table 4: Prompt examples for our experiments. Note that all "*[IMG*]*" are replaced with actual images during inference.

| Experiment | Prompt Example |
|---|---|
| Symmetrical few-shot CoT | You are given a puzzle. The puzzle features a set of visual patterns arranged in a matrix on the top, with the bottom right piece missing and six options at the bottom (marked by 1, 2, 3, 4, 5, or 6). Which option (either 1, 2, 3, 4, 5, or 6) fills the missing piece best? 
 *[IMG1]* 
 To solve this puzzle, we need to identify a pattern or rule that applies to the rows or columns of the matrix. Let's examine the rows and columns to see if we can discern any patterns. Looking at the first row, we see a yellow percentage sign followed by a yellow percentage sign. Hence, nothing changes in the row, moving from left to right. In the second row, there's a yellow percentage sign. Following the above pattern, we deduce that the missing piece in the second row is a yellow percentage sign. Now, let's look at the columns. The first column has a yellow percentage sign, followed by a yellow percentage sign. Hence, nothing changes in the column, moving from top to bottom. The second column has a yellow percentage sign. Following the above pattern, the missing piece in the second column must be a yellow percentage sign. Combining the observations from rows and columns, we conclude that the missing piece is a yellow percentage sign. 
 The provided options at the bottom are as follows: 
 1. Yellow percentage sign 
 2. Yellow plus sign 
 3. Two yellow circles 
 4. One yellow circle 
 5. Yellow division sign 
 6. Yellow cross 
 Given these options and our conclusion, option 1 fits our criteria and best fills the missing piece. 
 *[IMG2]* |
| Asymmetrical few-shot CoT | You are given a puzzle. The puzzle features a set of visual patterns arranged in a matrix on the top, with the bottom right piece missing and six options at the bottom (marked by 1, 2, 3, 4, 5, or 6). Which option (either 1, 2, 3, 4, 5, or 6) fills the missing piece best? 
 Demonstration 1: 
 Puzzle: 
 [[yellow percentage sign, yellow percentage sign], 
 [yellow percentage sign, ?]] 
 Options: 
 1: yellow percentage sign 
 2: yellow plus sign 
 3: two yellow circles 
 4: one yellow circle 
 5: yellow division sign 
 6: yellow cross 
 To solve this puzzle, we need to identify a pattern or rule that applies to the rows or columns of the matrix. Let's examine the rows and columns to see if we can discern any patterns. Looking at the first row, we see a yellow percentage sign followed by a yellow percentage sign. Hence, nothing changes in the row, moving from left to right. In the second row, there's a yellow percentage sign. Following the above pattern, we deduce that the missing piece in the second row is a yellow percentage sign. Now, let's look at the columns. The first column has a yellow percentage sign, followed by a yellow percentage sign. Hence, nothing changes in the column, moving from top to bottom. The second column has a yellow percentage sign. Following the above pattern, the missing piece in the second column must be a yellow percentage sign. Combining the observations from rows and columns, we conclude that the missing piece is a yellow percentage sign. 
 The provided options at the bottom are as follows: 
 1. Yellow percentage sign 
 2. Yellow plus sign 
 3. Two yellow circles 
 4. One yellow circle 
 5. Yellow division sign 
 6. Yellow cross 
 Given these options and our conclusion, option 1 fits our criteria and best fills the missing piece. 
 Let's solve the puzzle in the image, step by step, similar to the demonstrations. 
 *[IMG]* |

Table 5: Prompt examples for our experiments (Continued). Note that all "*[IMG*]*" are replaced with actual images during inference.

