# OpenReview forum: "The Curious Case of Nonverbal Abstract Reasoning with Multi-Modal Large Language Models"
_colmweb.org/COLM/2024/Conference — COLM_

### Official Review · Reviewer_G9SG · 2024-05-10

**Rating:** 6
**Confidence:** 3
**Ethics Flag:** 1

**Summary:**

This work focuses on evaluating MLLMs on `nonverbal abstract reasoning` tasks. The work reveals that the open-sourced models perform poorly on this set of tasks.

**Reasons To Accept:**

- The work reveals a major limitation of the current open-sourced MLLMs, leaving room for exploration and improvement in this area for future work.

**Reasons To Reject:**

- Many MLLMs are considered obsolete, given the rapid pace of advancement in this field. For example, the BLIP-2 series is not instruction-tuned. A more appropriate comparison could be made with Instruct-BLIP.
- The performance of `nonverbal abstract reasoning` tasks could heavily rely on image resolution. Evaluating MLLMs with higher resolutions (such as LLaVA-Next, VILA-1.5) may be necessary for further investigation, although these models have only recently been introduced.
- I would suggest adding experiments to ablate (1) the VIT visual encoder (to investigate the impact of higher resolution) and (2) the LLM decoders (to determine which part of MLLMs is more crucial for this task). This would help identify whether the bottleneck lies in visual perception or if performance is bounded by the reasoning ability of LLMs (such as in models 7b, 13b, 34b, or 70b).

---

> ### Author Rebuttal · Authors · 2024-05-26
>
> We thank the reviewer for their insightful feedback. In the following, we address their questions and concerns:
>
> **Q1:**
> At the time of submission, we used most of the available state-of-the-art models, including Instruct-BLIP, for our evaluations; however, as you mentioned, keeping up with the rapid pace of changes is costly.
>
> **Q2:**
> Similar to the previous questions, most of these models were not released during our evaluations. Our only examples of high-res models are Gemini and GPT-4v, which are closed-source. However, it’s important to note that the puzzles we used in our evaluations involve extremely simple shapes, such as squares or circles, with simple features like blue or yellow colors that do not require heavy reliance on detecting small details in a high-resolution image. Moreover, many models performed well on our visual awareness tests (Section 5, Figure 4), showcasing their understanding of the puzzles' details from a purely visual standpoint. At the same time, many of them performed extremely poorly when provided with all the details in a textual format (Section 5, Figure 3), which showcases a gap in reasoning abilities rather than perception abilities.
>
> **Q3:**
> We have done experiments on both vision-only (Section 5, Figure 4) and text-only variations (Section 5, Figure 3) as detailed in Section 5. However, given the focus on MLLM, we kept the model pool to existing MLLMs so as not to diverge from the paper's goal.

---

> > ### Author Response · Authors · 2024-06-04
> >
> > Dear Reviewer G9SG,
> >
> > Thank you again for your time. As we approach the end of the discussion period, we believe our response has addressed your questions and concerns. Please let us know if you have any further concerns or questions; we will be more than happy to address them.
> >
> > Best

---

### Official Review · Reviewer_kAJ8 · 2024-05-11

**Rating:** 6
**Confidence:** 4
**Ethics Flag:** 1

**Summary:**

This paper aims to study abstract reasoning of multimodal LLMs, more specifically the nonverbal abstract reasoning abilities of open-source and closed-source MLLMs using variations of Raven’s Progressive Matrices. The results are mostly negative, showing low performances, with some exceptions is some tasks solved by closed-source models. Chain-of-thought prompting is shown to be useful, as found in the literature.

**Questions To Authors:**

Please see my question above. They all request more information on how the assessments were done, on the motivations of the procedures for assessment.  This information and level of detail seems to be crucial given the goals of the paper.

**Reasons To Accept:**

I think this is an important area of investigation and the paper makes a good attempt to explore some of this space.

**Reasons To Reject:**

I don’t think this paper is convincing. Since it is mostly about negative results, we need to be reassured that the process was correct and this is a true negative result, and we are not.

- First, the datasets are visual and the textual part is simply a description of the visual input. So, it is really not clear why the authors expect MLLM to work on this kind of datasets, since the text part does not reproduce the abstract reasoning structure of the visual part.
                   Q1 What is the reasoning you have applied to think that this kind of data would reveal abstract analogical reasoning?

- Second, the paper is imprecise in its definition of the problem (what exactly constitutes nonverbal reasoning?) and methods used to assess the MLLM. Specifically, the non-computational aspects are  ill-defined and are missing crucial information for reproducibility and assessment of work.

- Q2 What are the qualifications of the result validators and how was their assessment validated?
- Q3 Why do you state that these tasks are  easy for humans?
- Q4 How do you define and quantify correct reasoning?
- Q5 How do you perform Inter-annotator Agreement?

---

> ### Author Rebuttal · Authors · 2024-05-26
>
> We thank the reviewer for their insightful feedback. In the following, we address their questions and concerns:
>
> **Q1 & Q2:**
> We used the term "nonverbal abstract reasoning," as defined by Huang et al. [1] (Section 4.2), referring to the task of providing IQ-style puzzles as images paired with short textual instructions of desired output. Moreover, Huang et al. [1] and Zhao et al. [2] provide evidence of non-trivial performance for such models. Due to our skepticism about the correctness of the underlying reasonings, we investigated this task using a much broader set of models and experimental questions in this work. Moreover, we used labor-intensive manual assessment to provide more grounded insights compared to automatic scores that don't capture the nuances of LLM-generated responses.
>
> To solve these puzzles, a model must first correctly detect the relations between the presented objects (e.g., a red circle turning into a blue circle in the first row) and then apply them to determine the missing piece (e.g., what would a red square turn into in the second row?), which requires a certain level of analogical reasoning. To the best of our knowledge, this is the first analysis of MLLMs on this task. We will publicly release our codes, datasets, annotations, and assessments to mitigate any concern over reproducibility or annotation quality.
>
> [1] Huang, Shaohan, et al. "Language is not all you need: Aligning perception with language models," NeurIPS 2023
>
> [2] Zhao, Haozhe, et al. "MMICL: Empowering Vision-language Model with Multi-Modal In-Context Learning," ICLR 2024
>
> **Q3:**
> All evaluators and annotators were mid-level to senior Computer Science PhD students specializing in NLP with extensive experience evaluating LLMs. Moreover, each sample was annotated by one person, and any uncertain case was flagged and shared among all three evaluators for discussion. After discussions, the final label was determined by a majority vote.
>
> **Q4:**
> We ran a study with 25 college-level participants with diverse backgrounds. Each participant was given ten randomly selected samples from the IQ50 dataset (20% of the dataset) and instructed to provide a short reason for their answer. The average performance of this group was 95.9%, with a 6.92% standard deviation. We will add these results to the paper to emphasize the performance disparity.
>
> **Q5 & Q6:**
> *Due to the rebuttal's limited space (2500 characters), we will provide more details once the discussion period begins.*

---

> > ### Author Response · Authors · 2024-06-04
> >
> > Dear Reviewer kAJ8,
> >
> > Thank you again for your time. As we approach the end of the discussion period, we believe our response has addressed your questions and concerns. Please let us know if you have any further concerns or questions; we will be more than happy to address them.
> >
> > Best

---

> > > ### Comment · Reviewer_kAJ8 · 2024-06-05
> > > **reply to author's response**
> > >
> > > Thanks for your convincing clarifications, I think it is important you find a way of including both the explanation of the motivation and a much more detail description of the assessment procedure in the final paper. I assume you will, and I have changed my score accordingly.

---

> ### Author Response · Authors · 2024-05-31
>
> Thank you once again for your insightful review. In a general comment, we have provided more details regarding human performance and the manual evaluation process. If you have any other questions or concerns, please feel free to let us know.

---

### Official Review · Reviewer_QZHA · 2024-05-11

**Rating:** 5
**Confidence:** 4
**Ethics Flag:** 1

**Summary:**

This paper presents a comprehensive evaluation of the nonverbal abstract reasoning abilities of multi-modal large language models (MLLMs) using variations of Raven's Progressive Matrices (RPM) as benchmarks. The authors test a wide range of open-source and closed-source MLLMs, providing insights into their current capabilities and limitations. The experiments reveal that while closed-source models like GPT-4V demonstrate non-trivial reasoning abilities, open-source models generally struggle with these tasks. The paper also uncovers critical shortcomings in both the visual and textual reasoning capabilities of MLLMs, setting performance ceilings. Interestingly, the authors find that techniques like Chain-of-Thought prompting and corrective hints can significantly boost performance, especially for closed-source models.

**Reasons To Accept:**

- The paper provides a thorough evaluation of a wide range of open-source and closed-source MLLMs on nonverbal abstract reasoning tasks. The authors use variations of Raven's Progressive Matrices as benchmarks, which are well-suited for assessing abstract reasoning abilities. The extensive experiments offer good insights into the current state-of-the-art in MLLM reasoning.
- The paper uncovers several interesting findings, such as the  performance gap between open-source and closed-source models, the critical shortcomings in both visual and textual reasoning capabilities, and the effectiveness of techniques like Chain-of-Thought prompting and corrective hints in boosting performance. These findings contribute to a better understanding of MLLMs and their limitations.

**Reasons To Reject:**

- The use of Raven's Progressive Matrices and techniques like Chain-of-Thought prompting have been explored in previous studies, albeit not as extensively as in this work.
- The paper compares a wide range of MLLMs but does not provide a detailed analysis of how the architectural differences between these models might influence their reasoning capabilities. A more thorough investigation of the relationship between model architecture and performance on abstract reasoning tasks could have provided additional insights and made the contributions more substantial.
- The authors acknowledge that the superior performance of closed-source models could be due to the nature of their training data. However, the paper does not delve into this aspect in detail. How fine-tuning on these tasks improve the open-source models' performance? A more comprehensive analysis of the potential impact of training data on the models' reasoning abilities would have strengthened the study's findings and implications.
- The paper compares the performance of MLLMs against random and majority baselines but does not include a human performance baseline. Including human performance on the same tasks would have provided a more informative context for interpreting the models' reasoning capabilities and limitations.
- The authors employ a manual inspection process to assess the reasoning abilities of the models, there may be concerns about the scalability and reproducibility of this approach. Providing more details on the rubric and the inter-annotator agreement could have addressed these concerns and increased the reliability of the manual evaluation results.

---

> ### Author Rebuttal · Authors · 2024-05-26
>
> We thank the reviewer for their insightful feedback. In the following, we address their questions and concerns:
>
> **Q1:**
> While we acknowledge that RPM and CoT are not new methods, our paper is more geared toward providing novel analysis and insight into MLLMs and their potential shortcomings rather than introducing novel approaches.
>
> **Q2 & Q3:**
> We agree that there is a need for further investigations on the effect of model architectures, training data, training regimens, fine-tuning, etc.; however, given the space of possibilities and cost of training these large models, such a study is prohibitively costly and broad for one article. Moreover, many of these details are often not disclosed publicly for closed-source models. As a result, we focused on testing a wide range of models (implicitly diversifying the approaches) in common zero/few-shot scenarios, providing novel insights into their strengths and limitations. We hope this work emphasizes the importance of these open challenges in future works.
>
> **Q4:**
> We ran a study with 25 college-level participants with diverse backgrounds. Each participant was given ten randomly selected samples from the IQ50 dataset (20% of the dataset) and instructed to provide a short reason for their answer. The average performance of this group was 95.9%, with a 6.92% standard deviation. We will add these results to the paper to emphasize the performance disparity.
>
> **Q5:**
> Regarding the rubric and the inter-annotator agreement, our main challenge was to overcome the nuances that appear in generated reasonings. As such, the evaluators first met to 1) determine the correct reasoning paths for the samples and 2) review examples of generated responses (e.g., GPT-4v, Gemini, etc.) to determine the evaluation strategy. Based on the observations in this initial meeting, the evaluators decided to allow for extra/wrong details in the generated responses as long as they did not affect or interfere with the alignment of the generated response to correct reasoning paths.
>
> For example, in some instances, the models perceived shadows (or incorrect colors) in the shapes that were not present in the puzzle; however, as long as they correctly detected the row-wandd/or column-wise change of patterns (e.g., square turning to circle) and grounded their reasoning on them, the evaluators marked the response as correct.
>
> *Due to the rebuttal's limited space (2500 characters), we will provide more details once the discussion period begins.*

---

> > ### Author Response · Authors · 2024-06-04
> >
> > Dear Reviewer QZHA,
> >
> > Thank you again for your time. As we approach the end of the discussion period, we believe our response has addressed your questions and concerns. Please let us know if you have any further concerns or questions; we will be more than happy to address them.
> >
> > Best

---

> ### Author Response · Authors · 2024-05-31
>
> Thank you once again for your insightful review. In a general comment, we have provided more details regarding human performance and the manual evaluation process. If you have any other questions or concerns, please feel free to let us know.

---

### Official Review · Reviewer_XKpq · 2024-05-14

**Rating:** 6
**Confidence:** 4
**Ethics Flag:** 1

**Summary:**

This study explores the nonverbal abstract reasoning capabilities of multi-modal large language models (MLLMs) using variations of Raven's Progressive Matrices, a benchmark traditionally used to assess human cognitive abilities. The research aims to bridge the gap in understanding the complex reasoning abilities of MLLMs, which integrate both verbal and visual information. Through rigorous testing, the study reveals significant performance disparities between open-source and closed-source MLLMs and identifies limitations in their visual and textual processing capabilities. The authors also investigate various enhancement strategies, notably Chain-of-Thought prompting, which dramatically improves the models' reasoning performance.

**Questions To Authors:**

Will there be copy right issues of using these data?

**Reasons To Accept:**

1. Timely topic.
2. Rich analysis and results.
3. Help to accelerate future research.

**Reasons To Reject:**

1. The reviewers has some concerns on the current experiment settings.

a. The impact of resolution and position representation. Current set up doesn't ablate much on whether the limited performance especially on zero-shot comes from the fact that the model is more suitable for other representation of locations and resolution. For example, is it possible that certain issue comes from the resolution/scale of the main object in the image? Is it possible that it is better to use language description of the locations such as top right, top left, bottom right and bottom right to let the model better understand the relative positions?

b. The manual evaluation lacks more rigorous analysis. There are only 3 annotators and the rubric is vague on the hallucination part. It is also unclear about inter-annotator agreement on the annotation results and the current setup makes it almost impossible to reproduce the results on one of the most important parts of the work: reasoning quality. It is also worthy to set up an automatic score for evaluation of reasoning process.

---

> ### Author Rebuttal · Authors · 2024-05-26
>
> We thank the reviewer for their insightful feedback. In the following, we address their questions and concerns:
>
> **Q1:**
> Regarding resolution, all samples were high-resolution (> 1000x600 pixels) to ensure all details were adequately presented. We will add this information to the paper for clarification. Regarding the objects' position, while it is true that each model has a slightly different preference, we tested them on the widespread use case of one picture with details centered from both directions. It is important to note that many models performed well on our visual awareness tests (Section 5, Figure 4), showcasing their understanding of the puzzles' details from a purely visual standpoint. At the same time, many performed exceptionally poorly when provided with all the details in a textual format (Section 5, Figure 3), showcasing a gap in reasoning abilities rather than perception abilities.
>
> Regarding the effect of textual description, our experimental goal was to test the current state of the models in a natural scenario where the user does not need to guide the model. In parallel, we have a set of experiments in Section 6.1 (Figure 5) to show how much we can boost their performance through targeted textual guidance.
>
> In general, we assumed that any well-trained model must work well in typical scenarios, as it would be challenging and costly for a user to find the optimal setting.
>
> **Q2:**
> Regarding the demographic and expertise of the evaluators and annotators, all were mid-level to senior Computer Science PhD students specializing in NLP with extensive experience working with and evaluating LLMs.
>
> Regarding the rubric and the inter-annotator agreement, our main challenge was to overcome the nuances that appear in generated reasonings. As such, the evaluators first met to 1) determine the correct reasoning paths for the samples and 2) review examples of generated responses (e.g., GPT-4v, Gemini, etc.) to determine the evaluation strategy. Based on the observations in this initial meeting, the evaluators decided to allow for extra/wrong details in the generated responses as long as they did not affect or interfere with the alignment of the generated response to correct reasoning paths.
>
> *Due to the rebuttal's limited space (2500 characters), we will provide more details once the discussion period begins.*
>
> **Q3:**
> No, the non-annotated version of the datasets has already been used in previous publications and is publicly available.

---

> > ### Author Response · Authors · 2024-06-04
> >
> > Dear Reviewer XKpq,
> >
> > Thank you again for your time. As we approach the end of the discussion period, we believe our response has addressed your questions and concerns. Please let us know if you have any further concerns or questions; we will be more than happy to address them.
> >
> > Best

---

> ### Author Response · Authors · 2024-05-31
>
> Thank you once again for your insightful review. In a general comment, we have provided more details regarding human performance and the manual evaluation process. If you have any other questions or concerns, please feel free to let us know.

---

### Author Response · Authors · 2024-05-31
**Details of Manual Evaluation and Human Performance**

We thank the reviewers for their insightful feedback. We are encouraged that they agreed with us on the timely manner and importance of this topic (R1, R3) and have found our analysis to be rich (R1) and insightful (R2, R3), revealing significant shortcomings in the current open-source MLLMs (R2, R4) and enabling future research (R1, R4). In the following, we address their questions and concerns regarding human performance and manual evaluation:

**Human Performance:**
To address the questions regarding the comparison to human performance, we ran a study with 25 college-level participants from diverse educational and demographic backgrounds. We provided each participant with ten randomly selected samples from the IQ50 dataset (20% of the dataset) and instructed them to solve the puzzle and give a short reason for their answer. The average performance of this group was **95.9%**, with a standard deviation of 6.92%. We will add these results to the paper to further showcase the performance disparity.

**Manual Evaluation:**
Here, we provide more details to address the questions regarding the manual evaluation procedures:

a) Regarding the demographic and expertise of the evaluators and annotators, all were mid-level to senior Computer Science PhD students specializing in NLP with extensive experience working with and evaluating LLMs.

b) Regarding the rubric and the inter-annotator agreement, our main challenge was to overcome the nuances that appear in reasonings generated by LLMs. As such, the evaluators first met to 1) determine the correct reasoning paths for the samples in the dataset and 2) review a series of sample responses generated by the models (e.g., GPT-4v, Gemini, etc.) to determine the evaluation strategy. Based on the observations in this initial meeting, the evaluators decided to allow for extra/wrong details in the generated responses as long as they did not affect or interfere with the alignment of the generated response to correct reasoning paths.

For example, in some instances, the models perceived shadows (or incorrect colors) in the shapes that were not present in the puzzle; however, as long as they correctly detected the row-wise and column-wise change of patterns (e.g., square turning to circle) and grounded their reasoning on them, the evaluators marked the response as correct.

Moreover, each sample was annotated and assessed by one person, and any uncertain case was flagged and shared among all three evaluators for discussion. After discussions, the final label was determined by a majority vote (i.e., at least 2 out of 3).

c) Regarding the scalability of evaluations, while we acknowledge the difficulty of scaling such manual experiments, one of the main challenges of correctly assessing generated responses is that automatic metrics like ROUGE, BERTScore, etc., fall short of adequately evaluating the semantic nuances. Hence, we decided to go down the path of highly labor-intensive human expert evaluation to provide precise, concrete, and grounded insights.

While we acknowledge that manual evaluation for LLM-generated responses is nuanced and challenging, we tried to avoid any pitfall that could invalidate our results (e.g., using only expert PhD students for assessments and annotations). Furthermore, we will release the models' outputs and the evaluators' annotations to mitigate further concerns about the quality of the evaluations.

---

### Decision · Program_Chairs · 2024-07-10

**Decision:**

Accept

**Comment:**

This work tests the ability of vision-language models to solve raven progressive matrices puzzles. The authors reveal the difficulties that models encounter in this setup, that closed-book models do better, and that chain-of-thought prompting helps.

In general, reviewers were in agreement this is a thorough study that extends our understanding of the abilities and shortcomings of current models. There were questions on the experimental setup and the authors provided good detail on how annotation was done, who the annotators were, what is human performance, the focus on human-based metrics. etc.